# Coordinated human-exoskeleton locomotion emerges from regulating virtual energy

**Rezvan Nasiri[1], Hannah Dinovitzer[1], Nirosh Manohara[1], Arash Arami** [1,2]*

**1** Department of Mechanical and Mechatronics Engineering, University of Waterloo, Waterloo, ON, Canada,
**2** Toronto Rehabilitation Institute (KITE), University Health Network, Toronto, ON, Canada

\* arash.arami@uwaterloo.ca

**Data Availability Statement:** All data will be available after acceptance of the manuscript at this DOI: 10.6084/m9.figshare.24910953 Note that the DOI is reserved and currently has an embargo that will be removed upon the acceptance. Meanwhile,

## Abstract

Lower-limb exoskeletons have demonstrated great potential for gait rehabilitation in individuals with motor impairments; however, maintaining human-exoskeleton coordination remains a challenge. The coordination problem, referred to as any mismatch or asynchrony between the user's intended trajectories and exoskeleton desired trajectories, leads to sub-optimal gait performance, particularly for individuals with residual motor ability. Here, we investigate the virtual energy regulator (VER)'s ability to generate coordinated locomotion in lower limb exoskeleton. *Contribution:* (1) In this paper, we experimented VER on a group of nine healthy individuals at different speeds ($0.6m/s - 0.85m/s$) to study the resultant gait coordination and naturalness on a large group of users. (2) The resultant assisted gait is compared to the natural and passive (zero-torque exoskeleton) walking conditions in terms of muscle activities, kinematic, spatiotemporal and kinetic measures, and questionnaires. (3) Moreover, we presented the VER's convergence proof considering the user contribution to the gait and introduced a metric to measure the user's contribution to gait. (4) We also compared VER performance with the phase-based path controller in terms of muscle effort reduction and joint kinematics using three able-bodied individuals. *Results:* (1) The results from the VER demonstrate the emergence of natural, coordinated locomotion, resulting in an average muscle effort reduction ranging from 13.1% to 17.7% at different speeds compared to passive walking. (2) The results from VER revealed improvements in all indicators towards natural gait when compared to walking with a zero-torque exoskeleton, for instance, an enhancement in average knee extension ranging from 3.9 to 4.1 degrees. All indicators suggest that the VER preserves natural gait variability and user engagement in locomotion control. (3) Using VER also yields in 13.9%, 15.1%, and 7.0% average muscle effort reduction when compared to the phase-based path controller. (4) Finally, using our proposed metric, we demonstrated that the resultant locomotion limit cycle is a linear combination of human-intended limit cycle and the VER's limit cycle. These findings may have implications for understanding how the central nervous system controls our locomotion.

a private link to the data for Journal editor can be found here: https://figshare.com/s/a8317eba400070250bf5.

**Funding:** - AA received NSERC Discovery under Grants RGPIN-2018-04850 -AA received New Frontiers in Research Fund - Exploration under Grants NFRFE2018-01698 and NFRFE2022-620 - AA received John R. Evans Leaders Fund Canadian Foundation for Innovation - AA received Ontario Research Fund (ORF) - HD received scholarship NSERC CGS-M The funders had no role in study design, data collection and analysis, decision to publish, or preparation of the manuscript.

**Competing interests:** The authors have declared that no competing interests exist.

# 1 Introduction

Coordinated human-exoskeleton interaction is essential for maintaining active participation of the user in the motor task to engage their neuromuscular system and to boost their motor recovery [1–3]. Despite exoskeleton design and control having undergone advancements over the last decade, which includes powered and unpowered exoskeletons to reduce metabolic cost [4–6], lightweight exoskeletons [7, 8], exomuscle [9], and exosuits [10–12], the performance of the existing rehabilitation exoskeletons are still far from optimal due to the unresolved human-exoskeleton coordination problem at the controller level. The human-exoskeleton conflict can arise from improper assistive torque profiles, a mismatch between the exoskeleton reference trajectories and the user intended trajectories, and often a time offset or asynchrony between the user and the exoskeleton. This physical conflict is more pronounced in individuals with residual motor capacity, such as those with incomplete spinal cord injury (accounting for 70% of spinal cord injury survivors [13]) and stroke survivors (considering 50–80% of stroke survivors can regain their ability to walk [14]).

It has been shown that passive trajectory tracking controllers worsen human-exoskeleton coordination [15, 16]. As a result, these controllers may inhibit users' voluntary movements, reduce their active contribution to gait control and can undermine the task-driven plasticity [15, 16]. To resolve this issue, there have been attempts to improve human-robot coordination by adapting the exoskeleton feedforward torque control [17], adapting the reference trajectory based on interaction force minimization [1, 18, 19], encoding the reference trajectory into the dynamical equations of Central Pattern Generator (CPG) [20] or Dynamic Movement Primitive (DMP) [21]. Although these methods adapts the exoskeleton behavior, they impose new dynamics of trajectory adaptation, different than natural dynamics of walking, that can worsen the coordination.

Another approach to resolve human-exoskeleton coordination problems is to make the controller time-independent using spatial-temporal decoupling. The notion of time was removed in the controllers developed in [22–24] by controlling the user hip-knee kinematic over a desired path; path controller. To encourage a velocity within a desired range, they added a time-tunnel over the desired path which makes their controllers time dependent. To address encoding the desired velocity, a velocity vector field (flow controller) was also designed to control the velocity about the desired path [25], however, this controller cannot provide the assistive torque during stance phase which is a huge limitation for an assistive lower limb exoskeleton controller. To solve this issue in a later study Martinez et al. switched from flow controller to a different controller during the stance phase [26]. To improve the human-prosthetic knee coordination and human-exoskeleton interaction at hip, recently, reinforcement learning (RL) based control solutions were proposed [27, 28]. Additionally, path, flow, and the proposed RL controllers require the detection of the user's gait events and intended gait phase. There are other controllers that attain the time independency by dynamic compensation rather than kinetic or velocity control [29–31]; they are robust to gait events. For instance, [29] compensates weight using the notion of potential energy shaping. However, such controllers try to fully compensate the dynamics and neglect the user contribution to the gait; they are not assist as needed.

To obtain a coordinated gait for the human-exoskeleton system, mentioned controllers rely on user adaptation to the exoskeleton assistive torques, and in some cases the adaptation of the exoskeleton to user behavior. Although user adaptation is essential for achieving a coordinated gait [32, 33], it could be challenging for those users with reduced sensorimotor capacities. Adaptation of exoskeleton to user behavior requires an accurate decoding of the timing of human intended movements. Given gait variability, which can be intensified by an

impairment, intention decoding based on gait kinematics has not been successful so far [34–36] despite the recent achievements in real-time gait phase estimation [37–40].

We developed an alternative approach, called virtual energy regulator (VER), for lower limb exoskeleton control; the initial idea is introduced and simulated in [41], then its feasibility has been tested on one healthy participant in [42]. Unlike path and flow controllers which control hip-knee position over a desired path or velocity, VER attains time independence by controlling position-velocity of each joint over the desired limit cycle. Benefiting from natural human dynamics, VER regulates a norm of angular position and velocity, called the virtual energy, to assist the movement at each joint. The basic concept of VER and its comparison with trajectory tracking controllers are shown in Fig 1.

Here, we investigate VER's ability in generating human-exoskeleton coordinated locomotion by running experiments on individuals with no known disability. We assess the resultant gait using kinematic, kinetic, and electromyographic analysis as well as a questionnaire and evaluate its closeness to natural walking with no exoskeleton. Furthermore, we compare the VER performance in terms of gait kinematics and muscle activity reduction with a phase based path (PBP) controller [22], enhanced with a robust gait phase estimator [40], on three able-bodied individuals. Finally, we use a computational model to explore whether the control of gait by the central nervous system, while walking with VER-controlled exoskeleton, resembles a VER controller itself.

## 2 Mathematics

The details of VER and its proof of convergence to limit cycle can be found in [41]; here, we provided a concise mathematics of VER and an approximation of the resulting limit cycle of the interaction between the VER-controlled exoskeleton and human user.

### 2.1 VER mathematical foundation

Fig 1 illustrates the VER regulating the virtual energy, defined as a phase-dependent squared distance to a desired feasible limit cycle. This results in formation of a feasible limit cycle at each joint.

**Definition. 1** (Feasible Limit Cycle). *A feasible limit cycle for VER is clockwise, sufficiently smooth, and simple (not-self-crossing) closed curve such that its radius ($r_d$) is a function of phase ($\theta$); using $\theta$, VER can estimate the gait phase, see [42] and* S1 Appendix *for more information. It has the following properties*:

$$
\begin{aligned}
r_d &= \sqrt{x_d^2 + \dot{x}_d^2} = f(\theta), \ \theta \in [-\pi \ \pi] \\
\theta &= \{0, \pi, 2\pi\} \rightarrow \dot{x}_d = 0 \\
\forall \theta &\in (0 \ \pi) \quad \rightarrow \partial x_d / \partial \theta < 0, \ \dot{x}_d > 0 \\
\forall \theta &\in (-\pi \ 0) \rightarrow \partial x_d / \partial \theta > 0, \ \dot{x}_d < 0
\end{aligned}
\tag{1}
$$

*where $x_d$ and $\dot{x}_d$ are the desired position and velocity defined over the limit cycle. Refer to Fig K in* S1 Appendix *for examples of non feasible and feasible limit cycles. As a result of this definition, over one period of limit cycle, phase is a monotonically decreasing function of time.*

To generate the desired limit cycle at each joint, the instantaneous states are shifted w.r.t. the origin ($(x_s, \dot{x}_s) = (x, \dot{x}) - \vec{O}, \ (x, \dot{x}) \in \mathbb{R}^2$) and then projected into the polar coordinate as: $r = \sqrt{x_s^2 + \dot{x}_s^2}, \ \theta = \tan^{-1}(\dot{x}_s / x_s)$. Using the polar representation, controlling each joint on the desired limit cycle is equivalent to satisfying $r(\theta) \equiv r_d(\theta)$. However, to have a well-posed mathematics, we choose $r^2(\theta) \equiv r_d^2(\theta)$ instead and define the control objective for VER as

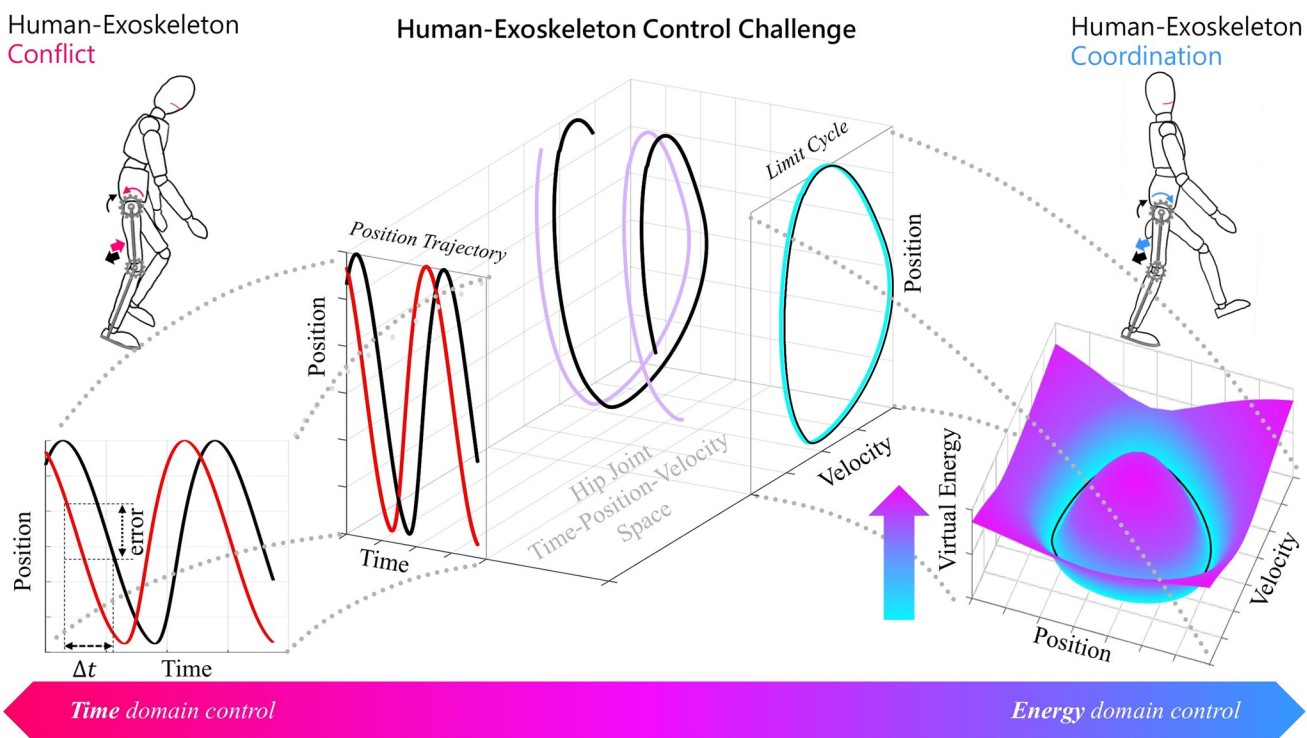

**Fig 1. The conceptual difference between conventional time-based controllers and virtual energy regulator.** Human-exoskeleton system dynamics at joint level can be expressed in the time-position-velocity space, two of these dimensions are sufficient to design a controller for such systems. Accordingly, three different scenarios can be imagined: i) position-time resulting in trajectory tracking controllers, ii) velocity-time resulting in velocity-based controllers, and iii) position-velocity resulting in limit cycle control (virtual energy regulator). The time-based controllers (i and ii) are sensitive to time offset between human intended and exoskeleton desired trajectories as they misinterpret a time difference as kinematic error (left figure) which can reduce human-exoskeleton coordination. In contrast, controlling in position-velocity domain (or virtual energy domain), allows time-independence and robustness to the time offsets, resulting in a better human-exoskeleton coordination (right figure).

$\Delta V = r^2(\theta) - r_d^2(\theta) \equiv 0$ where we call $V = r^2 = x_s^2 + \dot{x}_s^2$ the virtual energy as it captures aspects of kinetic energy of the system (due to $\dot{x}_s^2$) and potential energy of the system (due to $x_s^2$). In the following subsection, the VER controller ($\tau = \alpha + c + s$) is designed in terms of attractor ($\alpha$), compensator ($c$), and synchronizer ($s$) to regulate the virtual energy and generate the desired limit cycles across the joints.

## 2.2 VER attractor, compensator, and synchronizer

The VER controls each exoskeleton joint on a desired limit cycle by satisfying $\Delta V \equiv 0$. To achieve this objective, it computes the energy error ($\Delta V = V_d(\theta) - V(x, \dot{x})$), by comparing the virtual energy to the desired virtual energy; $V_d(\theta) = r_d^2(\theta) = x_d^2 + \dot{x}_d^2$. If the energy error is positive(negative), VER increases(decreases) the energy for which the applied torque/force should be in the same(opposite) direction as the joint velocity; see Fig K in S1 Appendix for illustration. Thus, the energy error should be multiplied by an odd function of velocity ($S(\dot{x}) \in \mathbb{R}$) which has the same sign as joint velocity; i.e., $S(\dot{x})\dot{x} > 0$. The knee joint velocity during stance phase is almost zero, hence to prevent chattering behavior for knee joint $S$ is set to $S(\dot{x}) = atan(\dot{x})$. The hip joint requires high torque about zero velocities during walking, hence to prevent torque drop by velocity reduction for hip joint $S$ is set to $S(\dot{x}) = sign(\dot{x})$. Therefore, the *VER attractor* term for $j$th joint can be obtained as $\alpha = S(\dot{x})P(V_d(\theta) - V(x, \dot{x}))$

where $P \in \mathbb{R}^+$ is a positive gain amplifying the deviations from the desired virtual energy. The *VER compensator*, which is a function of joint position and its time derivatives $(c(x, \dot{x}, \ddot{x}) \in \mathbb{R})$, compensates for undesired dynamics such as friction and Coriolis forces. The compensator is computed for each exoskeleton based on its identified dynamical model.

We define the limit cycle pitch ($\rho$) as the first harmonic phase of the limit cycle that is a function of limit cycle phase ($\theta$). To synchronize the limit cycles at the *i*th and *j*th joints with $\phi$ phase difference, we should satisfy $\rho_i(\theta_i) - \rho_j(\theta_j) \equiv \phi$ which can be written in polar coordinate as $\cos(\rho_i) \equiv \cos(\rho_j + \phi)$ and $\cos(\rho_j) \equiv \cos(\rho_i - \phi)$. To enforce such constraints, the *VER synchronizer* terms are implemented as $s_i = K(\cos(\rho_j + \phi) - \cos(\rho_i))$ and $s_j = K(\cos(\rho_i - \phi) - \cos(\rho_j))$ at the *i*th and *j*th joints, respectively; where $K$ is the synchronizer gain designed to provide stabilizing negative feedback for each joint towards the considered phase constraint. Synchronizers are designed to keep knee and hip joints of each leg in-phase ($\phi = 0$) while contra-lateral hip joints anti-phase ($\phi = \pi$). Adding the synchronizer term provides a global awareness across different joints; perturbations/disturbances in one joint lead to a proportional reaction in other joints, and a proper design of the phase difference across the joints grants mechanical stability in situations involving encountering obstacles or receiving bounded disturbances.

## 2.3 Human resultant limit cycle

To compute the resultant limit cycle by human contribution to the gait, consider the dynamical equation at each joint of human-exoskeleton system as

$$M\ddot{x} + h(\vec{x}, \dot{\vec{x}}, \ddot{\vec{x}}) = \alpha + c + u, \ h \in \mathbb{R} \tag{2}$$

where $x$, $\dot{x}$, and $\ddot{x}$ are the human-exoskeleton system joints' position, velocity, and acceleration. $M$ is the correspondence term of the system's mass matrix, $h()$ contains other dynamical terms at the targeted joint, and $u$ is the human contribution to the gait. In this case, we consider the virtual energy ($V$) as the Lyapanov function where its time derivative is $\dot{V} = (\partial V / \partial \dot{x})\ddot{x} + (\partial V / \partial x)\dot{x}$. By replacing $\ddot{x}$ from Eq 2 and substituting $\alpha = S(\dot{x})P(V_d - V)$, we have:

$$\dot{V} = \frac{2P}{M} S\dot{x}(V_d - V) + \frac{2}{M}\dot{x}(c - h + x) + \frac{2}{M}\dot{x}u. \tag{3}$$

Here we make assumptions, for the sake of argument, to achieve an approximation of resultant limit cycle. Note that later in the experimental results, we will come back to test the validity of these assumptions.

Let's assume that i) a compensator term ($c$) exists that satisfies $c = h - x$, ii) human contributions, i.e., torque profile, is the result of minimizing a virtual energy cost term by central nervous system, thus it can be written similar to VER attractor term as $u = RS(W_d - V)$ where $W_d$ is the human intended limit cycle and $R > 0$ is a proportional gain similar to $P$ in VER. Hence, we can rewrite Eq 3 as:

$$V = 2\frac{P + R}{M} S\dot{x}((1 - \beta)V_d + \beta W_d - V), \ \beta = \frac{R}{P + R}.$$

Based on this equation, for $V > (1 - \beta)V_d + \beta W_d$ we have $\dot{V} < 0$, and for $V < (1 - \beta)V_d + \beta W_d$ we have $\dot{V} > 0$; i.e., the gradient of limit cycle is convergent towards $V \equiv (1-\beta)V_d + \beta W_d$. Hence, based on *Poincare-Bendixson Criterion* [43], VER creates a stable limit-cycle behavior for each joint. In addition, the shape of the limit cycle is determined by $(1 - \beta)V_d + \beta W_d$ where $\beta$ represents as human contribution factor; $\beta$ shows how much human contributes to the

resultant gait. For instance, $\beta = 1(\beta = 0)$ indicates the resultant limit cycle is exactly the same as human-intended(VER-desired) limit cycle.

Using the emerged limit cycle $((1 - \beta)V_d + \beta W_d)$, we can infer that if a user's contribution to gait is minimal ($u \cong 0$), which due to limited motor capacity or resistance, then the resultant limit cycle will be the VER desired limit cycle. To test this extreme condition, we applied the VER controller to Indego exoskeleton without human subject. In this case, VER perfectly generates stable and synchronized desired limit cycles ($V_d$) across all controllable joints. This cannot be achieved using other controllers like path and flow controllers. The more the user contributes to the gait, considering their intended limit cycle being different than VER desired limit cycle, the emerged gait limit cycle would be closer to the human intended limit cycle. This was demonstrated in our experimental results.

### 2.4 VER limit cycle design

In most of the existing lower limb assistive exoskeletons, the ankle joint is passive [44] due to the cost efficiency, applicability, and safety issues [22]. Lack of actuation at the ankle joint in such exoskeletons results in walking pattern slightly different than natural walking [22]. In our controller, to maximize the similarity of the resultant gait to natural walking, the knee and hip limit cycles are designed similar but not identical to natural walking; see Fig 2a and 2d. This figure also compares the designed limit-cycles with their respective natural trajectories. VER requires feasible limit cycles (see section "VER mathematical foundation"), therefore, the knee desired trajectory/limit cycle needs to be modified during the stance phase. Accordingly, we removed the flexion behavior of the knee joint during the stance phase; hence, the controller encourages the participants to straighten their leg during the stance. It is important to note that modifying the desired motion profiles based on the exoskeleton and controller limitations is a common procedure in the design of lower limb assistive exoskeletons [44].

The desired limit cycle is simply the closed trajectory of the desired position and its corresponding velocity at each joint. The presented limit cycles are at the gait speed of $0.85 m/s$, and to generate the desired limit cycles at $0.75 m/s$ and $0.6 m/s$, the position is assumed to be the same and the velocity is scaled by 0.875 and 0.75.

### 2.5 VER torque and power

VER applies torque and power to each joint according to the position-velocity (state) of the joint in the phase plane; Fig 2e and 2f show the VER applied torques, and Fig 2g and 2h illustrate the distribution of VER power, respectively. The applied torque by VER attractor is $\alpha = S(\dot{x})P\Delta V$, and its gradient varies by two main variables: (1) joint state w.r.t. the desired limit cycle in sense of virtual energy ($\Delta V$: $\Delta V > 0$ and $\Delta V < 0$) mean the joint state is inside (lower than desired virtual energy) and outside (higher than desired virtual energy) of the limit cycle and (2) the joint velocity ($\dot{x}$). VER injects(dissipates) energy whenever the joint virtual energy ($V(x,\dot{x})$) is lower(higher) than the desired level defined over the desired limit cycle. Check Fig K in S1 Appendix for detailed information.

## 3 Experiment design

The experiments are designed to study the VER resultant gait's coordination and naturalness as two main concerns in lower limb rehabilitation. The experiments also compare the VER performance with the path controller which is a well-known and effective controller in the lower limb rehabilitation field and is tested on a large group of individuals with motor impairments [22–24]. Accordingly, experiment 1 is design to study the VER resultant gait on a large group of nine participants walking on the treadmill. In this experiment, we compare the

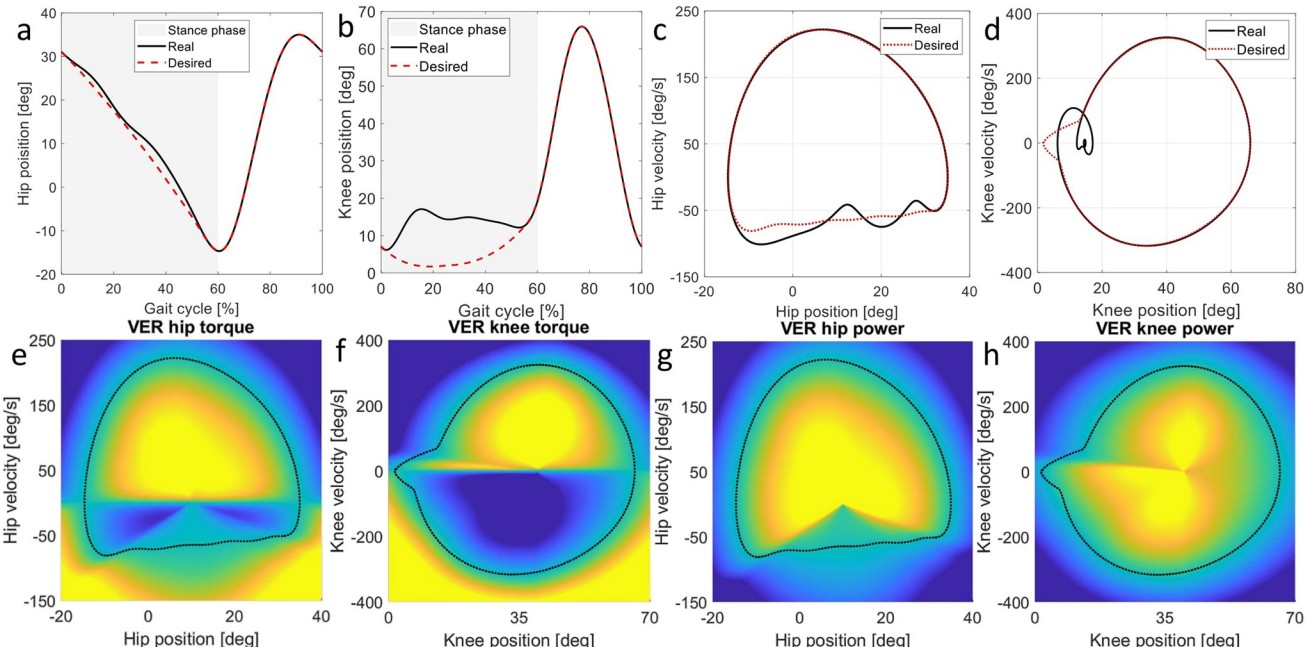

**Fig 2. Desired limit cycles and VER applied torque and power illustration.** (a-b) compare the natural and designed trajectories at the knee and hip joints, where the gray background indicates the stance phase [42]. (c-d) compare the natural and desired limit cycles at 0.85$m/s$ [42]. (e-h) illustrate VER applied torque and power w.r.t. the desired limit cycle. The color map shows the torque distribution w.r.t. the designed limit-cycles; yellow(dark blue) is positive (negative) highest value.

performance of walking on treadmill with VER controller, passive condition (zero-torque exoskeleton), and natural walking (no exoskeleton). The treadmill speed is also selected near to the walking speed suggested for individuals with motor impairments [22–28]. The experiment 2 is designed to compare the effect of VER controller and the phase-based path controller (PBP), equipped with a robust gait phase estimator [40], on treadmill walking performance. The PBP's hip and knee reference trajectories were selected according to the experimental (real) hip and knee trajectories shown in Fig 2a and 2b, respectively.

## 3.1 Experimental protocol

All participants were healthy with no known disability, and they provided written informed consent prior to the experiments. The study protocols and procedures were approved by the University of Waterloo, Clinical Research Ethics Committee (ORE 41794) and conformed with the Declaration of Helsinki. Similar to other studies [25], the whole experiment is done at speeds close to the preferable walking pace of individuals with motor impairments; i.e., 0.60$m/s$-0.85$m/s$. All participants attended two sessions: training and testing conducted on two consecutive days. During training, participants become familiarized with the exoskeleton through 20 minutes of overground and treadmill walking. The overground training was at self-selected pace while treadmill training involved speeds between 0.60$m/s$ and 0.85$m/s$. The testing sessions were different for each experiment.

**3.1.1 Experiment 1.** Nine participants (7 male and 2 female, age: 23.9±3.2 years, body mass: 72.3±7.0 kg, height: 177±6.3 cm; mean±standard deviation) participated in Experiment 1. In this experiment, testing comprised four consecutive trials with ten-minute rests between

trials. Participants filled out questionnaires after each trial. The first three trials include walking on the treadmill with exoskeleton at three different speeds ($0.60 m/s$, $0.75 m/s$, and $0.85 m/s$), for five minutes each. During each five-minute segment, there are three conditions: (1) Passive (Passive before) for two minutes, (2) Active VER for two minutes, and (3) Passive (Passive post) for one minute; Passive refers to the case that the exoskeleton is on (it is recording angles) but its applied torque is zero. The duration and order of phases at testing trials are designed to provide a comparative analysis between different conditions while preventing muscle fatigue. The last trial of testing sessions is treadmill walking without exoskeleton (Natural) with two minutes intervals at each speed ($0.6 m/s$, $0.75 m/s$, and $0.85 m/s$).

**3.1.2 Experiment 2.**   Three participants (2 male and 1 female, age: 23.5±5.2 years, body mass: 68.8±10.3 kg, height: 174±5.5 cm; mean±standard deviation) were participated in Experiment 2. In this experiment, testing comprised two consecutive trials of walking on the treadmill with exoskeleton at $0.8 m/s$ for two minutes rests between trials. During each trial, the exoskeleton is controlled on one of the controllers (VER or PBO), and the order of controllers are randomized for the participants.

## 3.2 Experimental setup

Fig 3 shows the experimental setup. The Indego explorer exoskeleton (Parker Hannifin, USA) with active knee and hip and locked-passive ankle joints is used. The exoskeleton measures joint positions, velocities, and motor torques at $200 Hz$.

After skin treatment, each participant is outfitted with 16 Trigno sEMG (Delsys, USA) acquiring the muscle activities of Tibialis Anterior (TA), Soleus (SOL), Gastrocnemius Medialis (GAS), Vastus Medialis (VAS), Rectus Femoris (RF), Tensor Fascia (TF), Biceps Femoris (BF), and Gluteus Maximus (GLU) of both legs at $2000 Hz$. Muscle activities were band passed ($25–500 Hz$), full-wave rectified and conditioned with a 100-sample moving average and normalized with maximum voluntary contraction (MVC). The 'muscle effort reduction' quantifies the decrease in muscle effort during Active condition compared to the Passive condition. Here, the muscle effort is determined by summing the cubed normalized muscle activity for each stride. The 'average muscle effort reduction' is the mean value of these individual 'muscle effort reductions'.

We use a split-belt instrumented treadmill (Bertec, USA), equipped with two 6 DoF force plates that measure the ground reaction force (GRF) and Center of Pressure (CoP) of each foot at $1000 Hz$. 16 reflective markers are placed according to the "PlugIn-Gait". An optoelectronic motion capture system containing eight Vero 2.2 cameras (Vicon, Motion System, UK) are used for collecting kinematic data at $100 Hz$.

# 4 Experimental result

## 4.1 Experiment 1

The VER performance in generating natural coordinated walking was evaluated using a thorough analysis of muscle activities, joint kinematics, limit cycles, toe clearance, and a questionnaire filled by the participants; ground reaction forces (GRF) were also recorded but presented in S1 Appendix. The presented graphs in coming subsections are mainly for $0.85 m/s$ and the results for $0.6 m/s$ and $0.75 m/s$ are reported in Figs E-G in S1 Appendix; the results show no significant difference across speeds.

**4.1.1 Muscle activity.**   Fig 4 presents a representative participant's muscle activities for eight different muscles under four conditions; 'Natural', 'Passive before', 'Active', and 'Passive post'. Note that the results of right and left legs are similar, hence, only the EMG results of the right leg are presented.

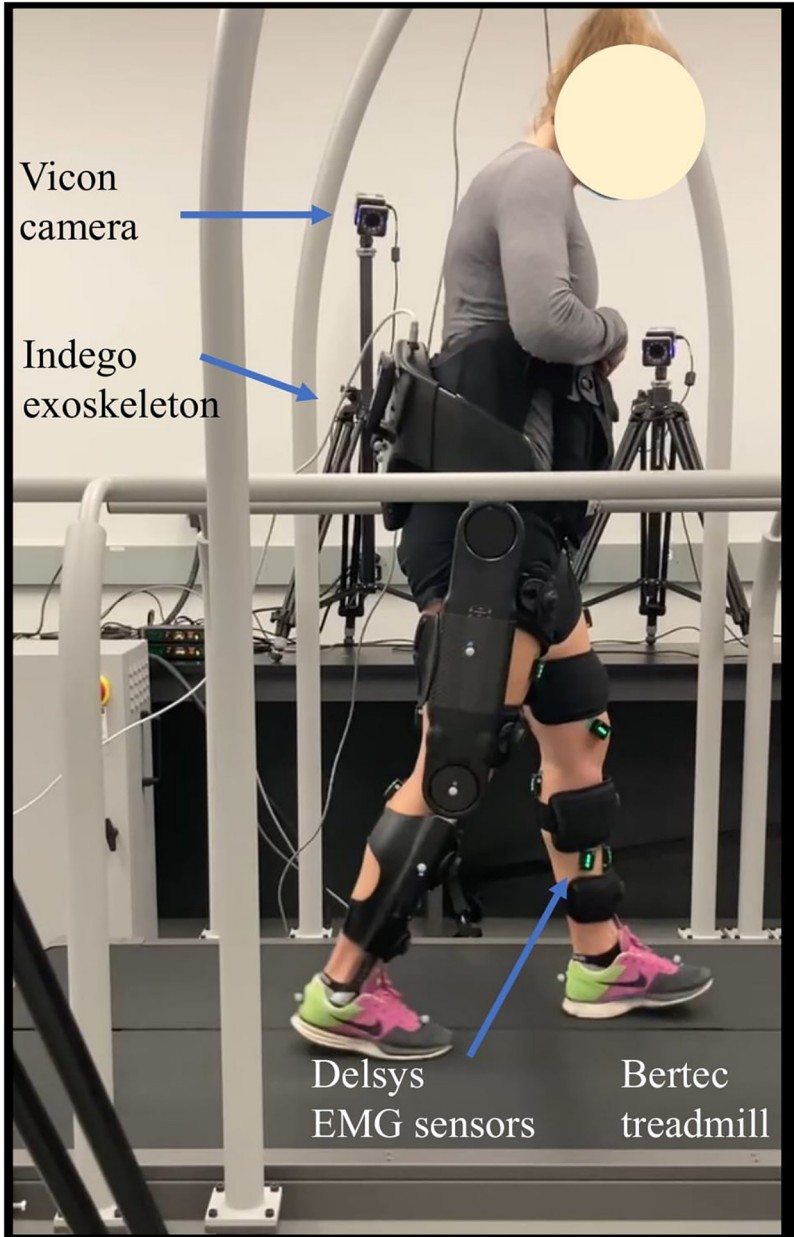

**Fig 3. The experimental setups.** The experimental setups, including Indego explorer exoskeleton (Indego, Parker, USA), 16 surface electromyography (sEMG) Trigno sensors (Delsys, USA), split-belt instrumented treadmill (Bertec, USA) equipped with two 6-degree of freedom (DoF) force plates, and optoelectronic motion capture system containing eight Vero 2.2 cameras (Vicon, Motion System, UK) 16 reflective markers are placed according to the "PlugIn-Gait" recommendation for lower body kinematic measurements.

According to Fig 4, SOL, TA, and GAS activation patterns are similar to the normal walking at $0.85m/s$. Similar patterns can also be seen at $0.6m/s$ and $0.75m/s$; see Figs A and B in S1 Appendix. However, BF and GLU activation patterns are different than those of natural walking; i.e., an extra peak occurs for BF(GLU) during stance(swing) compared to natural walking. According to the bar charts presented in Fig 4, with the exception of the activity of two muscles

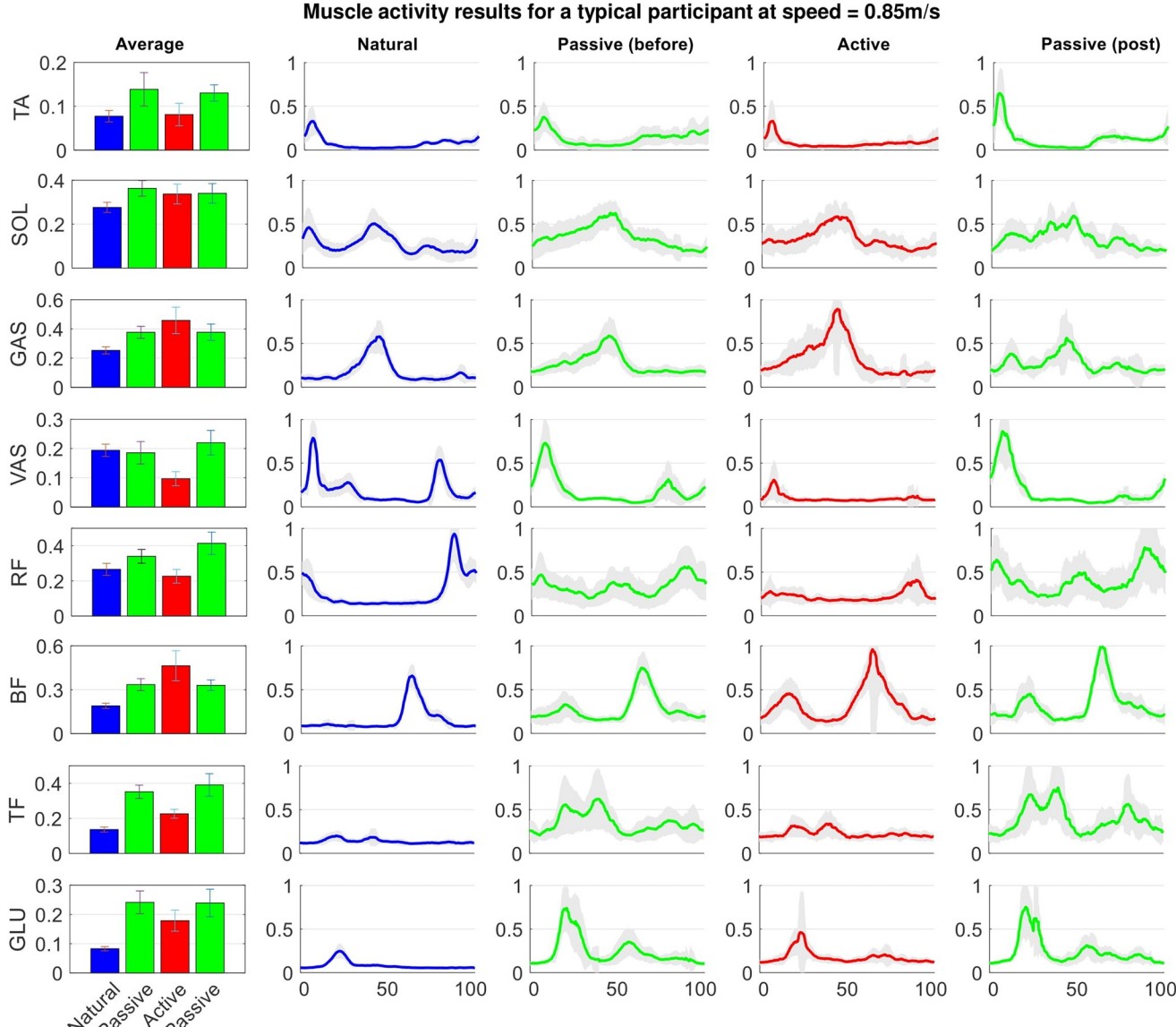

**Fig 4. Muscle activation pattern comparison at 0.85*m/s*.** Comparison between muscle activation patterns and average muscle activity for a representative participant in four different conditions; Natural, Passive before, Active, and Passive post. For this representative participant, VER (Active condition) results in average muscle effort reduction of 17.5% compared to Passive condition.

(BF and GAS), the activity of the rest shows a significant reduction when the controller turns on (Active) compared to the cases that the exoskeleton torque is zero (Passive). Among the six muscles with average activity reduction, VAS and RF average activities are even lower than walking without the exoskeleton; i.e., these muscles become silent when the controller turns on. The similar patterns can also be seen with the other speeds and in most of the participants (7 out of 9) so that VER achieves 'average muscle effort reductions' (refer to section "Experimental setup" for the definition) of 14.4% ± 4.0, 17.7% ± 6.2, and 13.1% ± 5.7 across all participants for 0.6*m/s*, 0.75*m/s*, and 0.85*m/s* speeds, respectively. The *p*-value for 'average muscle effort reductions' at each speed is *p* = 0.0156 (*two-sided Wilcoxon signed rank test*), which is

statistically significance with confidence interval of 95%. Note that although the variations seen across some of the results are not the same, similar *p*-values were obtained in some cases; this is due to the used *non-parametric Wilcoxon signed rank test*.

**4.1.2 Kinematics.** Fig 5a and 5h compare the desired trajectory and limit cycle with the Natural, Passive, and Active conditions for a representative participant and also across all participants at 0.85*m/s*. Fig 5i and 5j compare the Passive, Active, and Natural conditions in terms

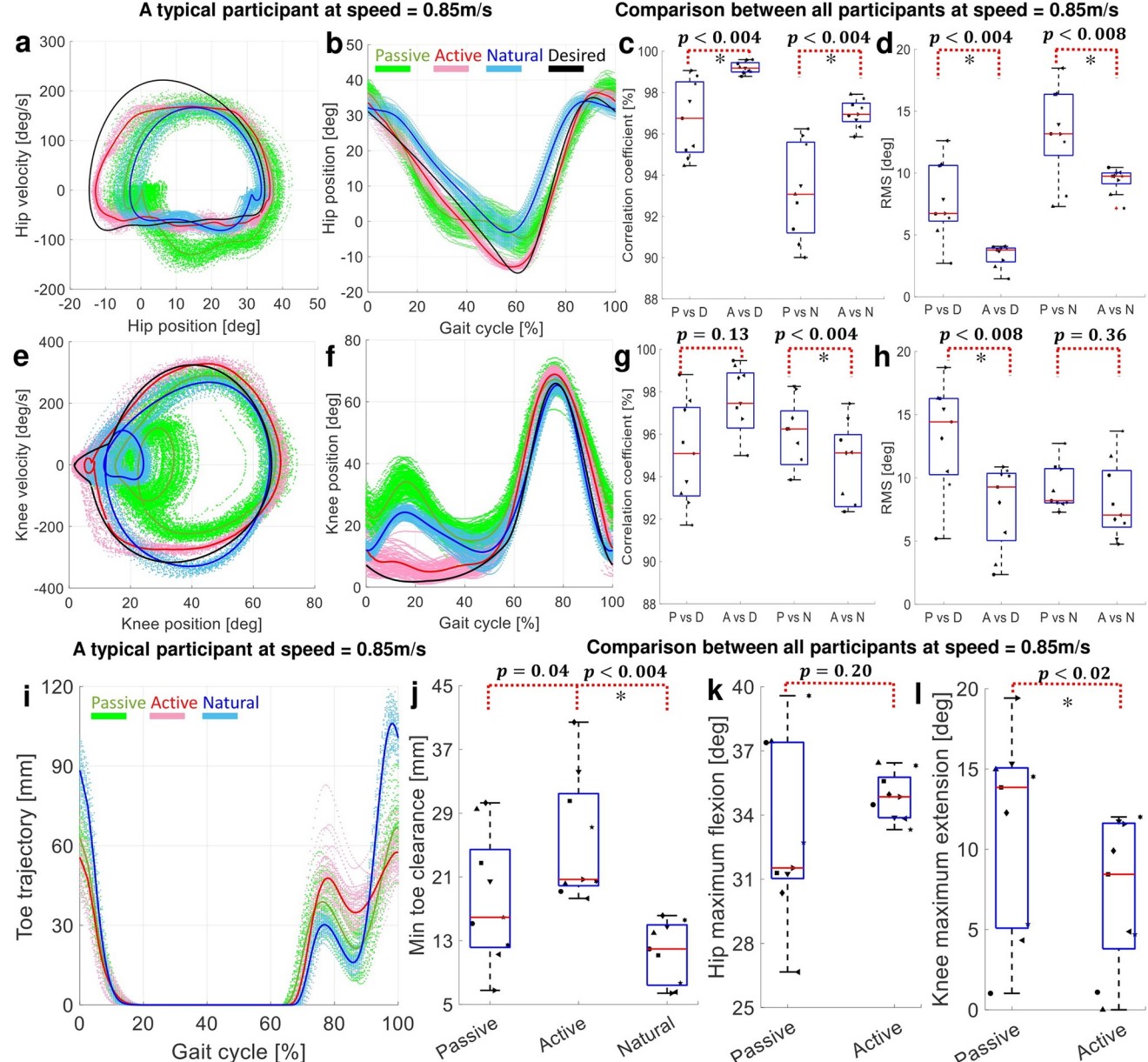

**Fig 5. The kinematic comparison at 0.85*m/s*.** (a,b,e,f) compare the Desired trajectory and limit cycle with Passive, Active, and Natural conditions at hip and knee joints for a representative participant. (c,d,g,h) compare the correlation coefficient and RMS of deviations from the Desired and Natural trajectories with Active and Passive conditions at hip and knee joints across all participants. (i) shows the toe clearance trajectory for a representative participant, and (j) compares minimum toe clearance of Passive, Active, and Natural conditions across all participants. (k,l) compare Passive and Active conditions in terms of hip(knee) maximum swing(stance) flexion(extension) across all participants; zero angle corresponds to a fully extended knee.

of toe clearance for a representative participant and across all participants at $0.85m/s$. In addition, Fig 5k and 5l compare the hip maximum swing flexion and knee maximum stance extension for Passive and Active conditions. The results for other speeds are reported in Figs C and D in S1 Appendix.

The exoskeleton mass, passive ankle joint, and its reduced degrees of freedom (constraining user's hip and knee to sagittal plane motion) all together contribute to the deviation of the participant gait from the Natural. This is evident in reduced range of motion (ROM) at knee and hip joints. Especially in the Passive condition, the knee joint does not extend fully during stance and the hip joint cannot fully flex during swing. During the Active condition, regulating the virtual energy at joints by VER results in trajectories more similar to the VER desired trajectory, and participants experience an improved knee extension during stance (maximum knee extension increased by 3.9 ± 4.7, 2.9 ± 5.1, and 4.1 ± 4.9 deg with $p$-value of 0.0195, 0.2031, and 0.0195 for $0.6m/s$, $0.75m/s$ and $0.85m/s$ speeds, respectively). VER also improved the hip maximum flexion during swing (by 2.6 ± 3.2, 2.5 ± 2.8, and 1.7 ± 3.6 deg with $p$-value of 0.0742, 0.0391, and 0.2031 for $0.6m/s$, $0.75m/s$ and $0.85m/s$ speeds, respectively); see Fig 5k and 5l.

The contribution of VER to the gait makes the hip and knee trajectories more similar to both Desired and Natural trajectories. To study this similarity, we compare Passive and Active with Desired and Natural trajectories in terms of correlation coefficient and RMS of error; see Fig 5c and 5d for hip joint and Fig 5g and 5h for knee joint. Comparing Passive and Active conditions, VER improves both indices at knee and hip joints, except for correlation coefficient at the knee joint when computed with respect to the Natural trajectory. This is due to the modifications applied to the knee desired limit cycle to satisfy the "feasible limit cycle" condition of the VER controller; please see section "VER limit cycle design" and Def. 1 in section "VER mathematical foundation" for more details. Although the presented results show the quality of the VER control performance to move the joint movements towards the Desired limit cycle, still the participant can experience a sufficient level of gait variability in Active condition indicating the user voluntary contribution to the gait.

Interestingly, the resultant limit cycles and trajectories in knee and hip fall between the Natural and Designed limit cycles and trajectories; this is analytically studied in section "Human resultant limit cycle" and discussed in section "Human contribution factor" that raises an interesting hypothesis that "human central nervous system may employ the same strategy as VER to control the lower limbs and generate stable limit cycles resulting in cyclic gait". As another consequence of the VER, the minimum toe clearance is increased for most of participants (8 out of 9) compared to Passive condition, which is due to improved knee extension and hip flexion; see Fig 5j. Particularly in the selected representative participant the toe clearance is increased by $13.6mm$ compared to Passive condition; see Fig 5i. Toe clearance is exaggerated in walking with the Indego exoskeleton (Passive and Active) compared to Natural condition. Indego has a passive ankle that limits the ankle ROM, hence the foot lifts off the ground faster than Normal and the toe maintains a higher distance w.r.t. the ground; see Fig 5i as a typical result.

**4.1.3 Human contribution factor.**   Our experimental results (see Fig 5) revealed that the emerged gait limit cycles, in pink, fall between the Desired VER limit cycle ($V_d$), in black, and those ($W_d$), in cyan, obtained during Natural condition (without exoskeleton). The VER forms a stable limit cycle that is a linear combination of the human intended limit cycle and the VER desired limit cycle ($V = (1 - \beta)V_d + \beta W_d$ where $0 < \beta < 1$); see section "Human resultant limit cycle" for an analytical derivation based on the assumption of human controller is VER-like. To test if the experimentally obtained limit cycles support this hypothesis, we solved linear regression problems to obtain $\beta$ for each participant at each joint. At $0.85m/s$, we obtained $\beta$

for different participants with the following $R^2$ values.

$$\begin{aligned}
\beta_{hip} &= (0.41, 0.66, 0.57, 0.62, 0.76, 0.41, 0.64, 0.50, 0.52) \\
\beta_{knee} &= (0.21, 0.07, 0.71, 0.83, 0.77, 1.29, 0.86, 0.68, 1.02) \\
R^2_{hip} &= 0.93 \pm 0.03 \qquad R^2_{knee} = 0.91 \pm 0.06
\end{aligned}$$

The obtained high $R^2$ and the $0 < \beta < 1$ at hip joint (also the case at other speeds) confirm that the emerged limit cycles at hip is a linear combination of the two limit cycles as predicted above. At knee joint, although two participants seem to overcompensate for VER desired limit cycle (indicated by the obtained values higher than one), the majority (7 out of 9) of participants showed $0 < \beta < 1$ with large $R^2$s corroborating to resultant interpolated limit cycles.

Obtained $0 < \beta < 1$ (using kinematic data in section "Kinematics") supported with average muscle activity reduction compared to Passive case (reported in section "Muscle activity") indicates that VER established a proper coordination between the human and exoskeleton. This allows participants to contribute to gait control with the exoskeleton. Even though this evidence does not prove that our central nervous system controls our lower limb during locomotion by regulating similar virtual energy term, it demonstrates the plausibility of this mathematical model for human gait control, warranting exploration in future research endeavors. This mathematical model can be compared in the future to the existing human gait control frameworks, for instance, those based on joint variable mechanical impedance [45–48], neuromuscular spinal reflexes [49], muscle synergy [50], and inverse optimal control [51].

**4.1.4 Control performance.** Fig 6a and 6b compare the Passive and Active virtual energies at hip and knee joints alongside their respective Desired profiles (i.e., radius of the limit cycle squared, against the limit cycle phase). The data pertains to a representative participant walking at a speed of $0.85 m/s$; the gray backgrounds indicate the stance phase.

As can be seen, Active condition regulates the virtual energy of the human-exoskeleton system closer to the desired virtual energy (black solid line). Particularly, hip virtual energy from limit cycle phase of $-180 deg$ to $-60 deg$ (corresponding to 0% to 40% gait cycle) and from $30 deg$ to $180 deg$ (corresponding to 60% to 100% of gait cycle) matches the desired virtual energy. Knee virtual energy got better regulated from limit cycle phase of $-90 deg$ to $20 deg$ (corresponding to 50% to 70% gait cycle) and from $150 deg$ to $180 deg$ (corresponding to 87% to 95% of gait cycle). Similar patterns can be seen at other speeds; see Figs H and I in S1 Appendix. Moreover, Fig 6c and 6d compare the virtual energy of Desired with Active and Passive conditions at hip and knee joints across the participants at $0.85 m/s$. The overall results show a statistically significant improvement at hip joint in terms of correlation coefficient and RMS comparing Passive and Active conditions with Desired profile. Although the VER regulates the virtual energy of hip and knee joints towards their Desired profiles, it does not enforce it strictly, allowing the participant to maintain a desirable degree of gait variability. Note that this natural gait variability does not associate with any increased risk of fall [52], and could contribute to an improved ability to recover perturbations, as it is in line with uncontrolled manifold theory [53].

**4.1.5 Questionnaire results.** A questionnaire is used to evaluate the subjective perception of safety, comfort, stability, effort and time to exhaustion/fatigue; for more details see S1 Appendix. Fig 6e and 6f compare our questionnaire results for Passive and Active conditions at $0.6 m/s$ and $0.85 m/s$; the questionnaire results for $0.75 m/s$ are reported in Fig J in S1 Appendix. In this graph, a score of 10 for safety, comfort, and stability means equivalent to the Natural condition and zero is the minimum possible score. For effort, score of zero means that participant walks with almost no effort and 10 means maximum possible effort. Finally, for the fatigue, we asked how long the participant can walk until feeling exhausted and its values are

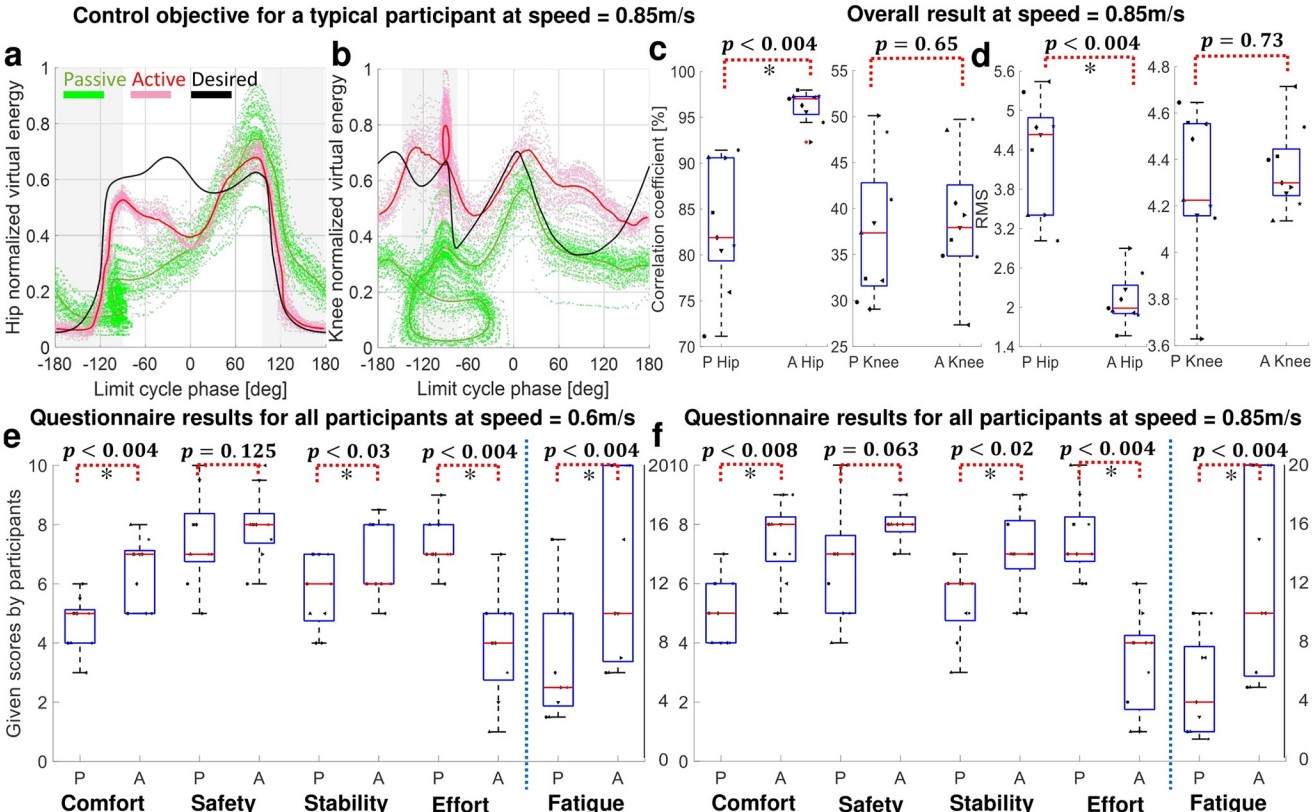

**Fig 6. VER control performance at 0.85$m/s$.** (a,b) describe virtual energy against the limit cycle phase where the gray backgrounds indicate the stance phase. (a,b) compare the desired virtual energy with virtual energy in two different conditions (Passive and Active) at hip and knee joints for a representative participant. (c,d) compare Pearson correlation coefficient and RMS of deviation from the desired virtual energy in two different conditions (Passive and Active) for hip and knee joints across all participants. (e,f) The questionnaire results for all participants at 0.6$m/s$ and 0.85$m/s$. The box plots compare Passive with Active condition in terms of comfort, safety, stability, effort, and time to fatigue. The vertical axes for time to fatigue is in right side of the plots.

shown in the most right box plots. According to the questionnaire scores, VER can improve all of the scores compared to Passive condition towards the Natural walking condition at all tested speeds; i.e., participants perceived a significant improvement in most of indices. While their medians have increased, safety score improvement was not statistically significant. The improvement of scores when VER is active is more pronounced at 0.85$m/s$ which is closer to the participants natural walking speed.

## 4.2 Experiment 2

The VER performance was compared with PBP in terms of muscle activity reduction and joint kinematics at 0.8$m/s$, and the results are illustrated in Fig 7. All participants acknowledged the satisfactory performance of both controllers, in terms of motion naturalness, comfort, time to fatigues, stability, and safety compared to Passive condition. Besides, they preferred the VER over the PBP controller for walking on treadmill at 0.8$m/s$. Fig 7a shows the muscle effort reduction for each muscle and participant in VER compared to PBP case, where the average muscle effort reduction for Participant 1, Participant 2, and Participant 3 are 13.9%, 15.1%, and 7.0%, respectively. Based on Fig 7a, VER leads to muscle effort reduction in VAS, BF, and GLU for all participants. It should also be noted that due to Indego passive ankle joint, muscle

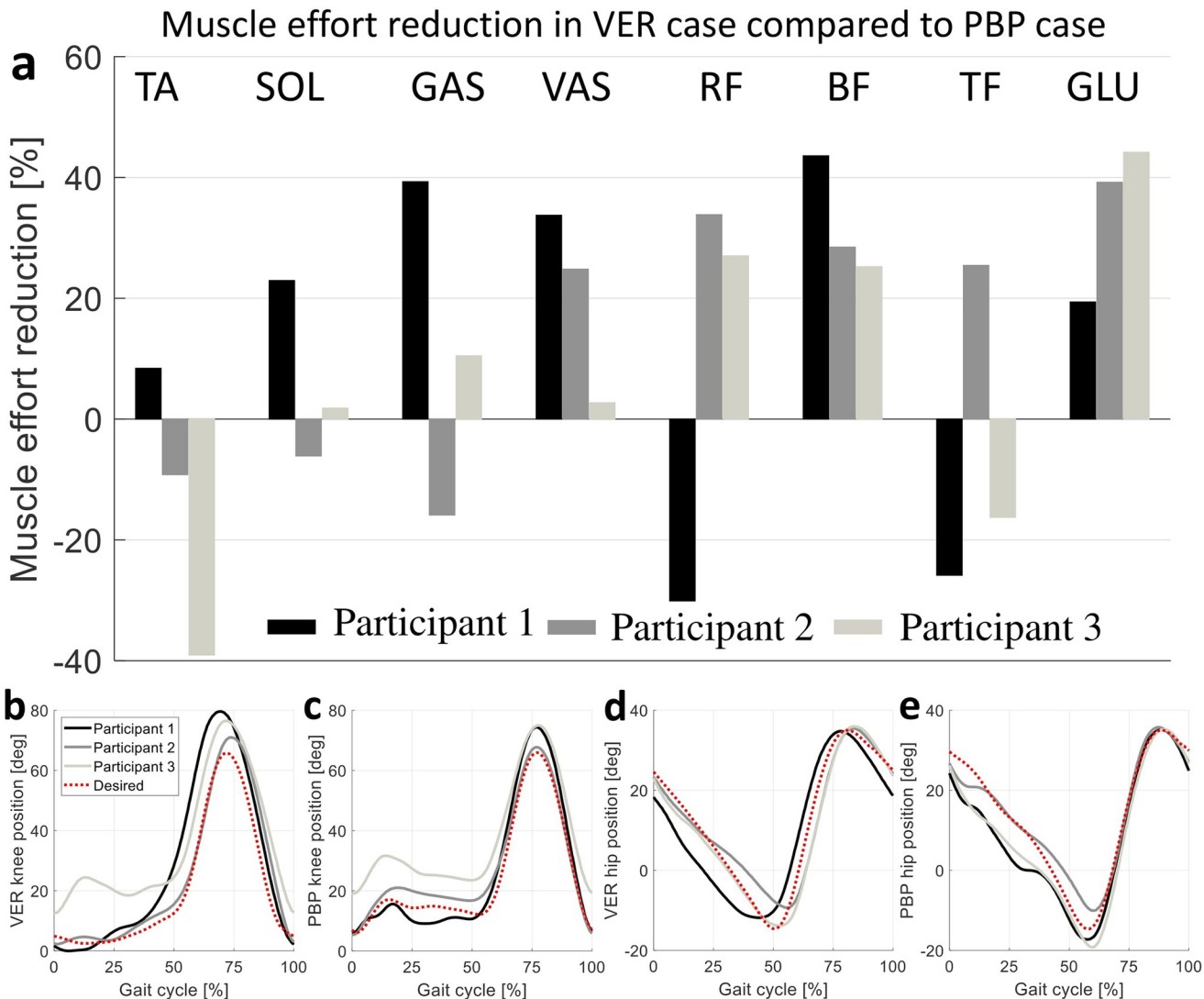

**Fig 7. Comparison between VER and PBP controllers at 0.8m/s.** (a) illustrates the muscle effort reduction (of eight different muscles) due to VER compared to the PBP case for three different participants; the positive values indicate VER-resultant relative muscle effort reduction. (b-e) compare the participants knee and hip joint kinematic when using VER and PBP controllers.

efforts of ankle mono-articular muscles (i.e., SOL and TA) mostly depend on the participant interaction with exoskeleton rather than the controllers performance. In total, an average muscle effort reduction of 12.0%±5.0% (average ± sem) across all participants was obtained with VER, which is statistically significant with confidence level of 95% (*two-sided Wilcoxon signed rank test p* = 0.0299).

According to Fig 7b and 7c, comparing the resultant knee trajectories of VER and PBP with their reference trajectories indicates a proper control performance for both VER and PBP controllers at the knee joint. However, the knee resultant trajectory during the stance phase for VER and PBP are different which is due to the modification of knee trajectory in VER during stance phase to make it a feasible limit cycle; see section "VER limit cycle design". It can also be seen that Participant 3 has a higher knee flexion during the stance phase for both controllers

compared to other participants which might be due to the different gait patterns of the Participant 3 during stance. Besides the stance phase, the VER provides a higher knee flexion during the swing phase which leads to a higher (about 9$mm$) toe clearance. Fig 7d and 7e show both controllers have a proper and similar tracking performance for the hip joint during stance and swing phases.

To sum up, (1) PBP and VER exhibit a similar kinematic and control performance, (2) subjective feedback of the participants favored VER over PBP, and (3) VER resulted in a significant average muscle effort reduction compared to PBP controller. These results demonstrate the superiority of VER as a time-independent controller compared to PBP (phase-based path controller enhanced robust gait phase estimator [40]) as a state of the art method.

## 5 Conclusion

In this paper, we implemented the VER, a new controller to resolve the gait coordination issue in lower limb rehabilitation, on an assistive limb exoskeleton and extensively tested it on nine able-bodied participants walking at three different speeds close to walking speed in impaired individuals. We analyzed the VER performance in terms of gait coordination and motion naturalness using muscle activity, motion kinematics, controller command, and questionnaires results. To investigate the advantages of the VER over the existing gait assistance controllers, the VER is also compared with a well-known and effective controller (for individuals with motor impairments) in the field; i.e., the path controller. Finally, we proved that the VER convergences to an intermediate limit cycle between the human intended and exoskeleton desired limit cycles. And, accordingly, a new metric is proposed to measure the VER contribution to the gait.

The results demonstrate that VER regulates the virtual energy at each joint towards the desired pattern resulting in a natural gait pattern with appropriate variability. The resulting natural variability in gait can be attributed to the correction of the sum of squared of joint angle and velocity by VER, rather than imposing constraints or correct each of these terms separately. The ease of maintaining a coordinated gait with VER controller was demonstrated by the decreased activities of most muscles when compared to walking with Passive condition; i.e., zero-torque exoskeleton. Besides, the total muscular efforts in VER controlled condition have reduced compared to Passive condition by 14.4% ± 4.0, 17.7% ± 6.2, and 13.1% ± 5.7 at 0.6$m/s$, 0.75$m/s$, and 0.85$m/s$, respectively. Nevertheless, the 'average muscle effort reduction' can be attributed more to exoskeleton dynamic compensation rather than coordination effect. The utilized exoskeleton has no active ankle joint resulting in an inability to compensate for a significant portion of dynamics. Moreover, observing no significant change in 'average muscle effort' compared to the Natural condition rejects this argument.

Passive walking alters the gait kinematics due to extra mass and inertia of the exoskeleton. VER however improves the reduced knee extension by 3.9 ± 4.7, 2.9 ± 5.1, and 4.1 ± 4.9 deg and hip flexion by 2.6 ± 3.2, 2.5 ± 2.8, and 1.7 ± 3.6 deg at 0.6$m/s$, 0.75$m/s$, and 0.85$m/s$, respectively. This increased ROM extends the usability of VER-controlled exoskeleton as a suitable tool for lower limb rehabilitation of individuals with residual motor functions, maintaining large ROM and volitional contribution of the users. VER also increases the minimum toe clearance of participants compared to both passive and natural (without) walking conditions. This latter could have a positive effect on the perceived stability of gait as portrayed in the questionnaire results.

The questionnaire results on perceived high comfort and stability, and low effort and fatigue also corroborate to the fact that such time-invariant controller can resolve the human-robot coordination problem. All participants found walking with exoskeleton during 'Passive

post' more difficult than 'Passive before' indicating the participant reliance on VER (shaped through user adaptation to VER) for walking. Comparison of the muscles' activation patterns, 'Passive before' and 'Passive post', showed no sign of fatigue or change in overall muscle activation patterns, which refutes the possibility of muscular fatigue being a factor in the difficulty perceived in 'Passive post' versus 'Passive before' condition.

As VER injects joint-level dynamics, compatible with dynamics of user's gait, no need for adaptation was perceived by the users. Conversely, a new gait dynamics emerges from the interaction of VER with the user as manifested by the resultant limit cycles falling in between the VER limit cycles and those intended by the user. Comparing the VER and PBP controllers showed that total muscular efforts in VER controlled condition have reduced compared to phase-based path controller by 13.9%, 15.1%, and 7.0% for three different participants at $0.8m/s$, respectively. In addition, compared to PBP controller, VER leads to a higher toe clearance ($9mm$), which provides a higher level of stability.

In conclusion, considering the participants feedback, having a similar muscle activation pattern compared to the Natural condition, and having a significantly lower 'average muscle effort' compared to the Passive condition and phase-based path controller indicate the VER coordinated behavior with participants' gait. These may attribute to a reduced human-exoskeleton physical conflict achieved by VER. While all the tests are done on able-bodied participants, we acknowledge that the selected speeds are lower than adult normal walking speed. This was done to ensure the effectiveness of VER in those speeds, with a future goal in mind to test VER on those speeds on participants with incomplete spinal cord injury. Additionally, in our future study, we aim to present a metric to measure human-exoskeleton coordination and evaluate the effectiveness of assistive controller, and test VER at different speeds and walking conditions.

## Supporting information

**S1 Appendix. Additional experimental results.**
(PDF)

**S1 Video. Experiment video file 1.**
(MP4)

**S2 Video. Experiment video file 2.**
(MP4)

**S3 Video. Experiment video file 3.**
(MP4)

## Author Contributions

**Conceptualization:** Arash Arami.

**Data curation:** Rezvan Nasiri, Hannah Dinovitzer, Nirosh Manohara.

**Formal analysis:** Rezvan Nasiri, Hannah Dinovitzer, Nirosh Manohara.

**Funding acquisition:** Arash Arami.

**Investigation:** Rezvan Nasiri, Hannah Dinovitzer, Arash Arami.

**Methodology:** Rezvan Nasiri, Arash Arami.

**Supervision:** Arash Arami.

**Validation:** Rezvan Nasiri.

**Visualization:** Rezvan Nasiri, Arash Arami.

**Writing – original draft:** Rezvan Nasiri, Hannah Dinovitzer, Arash Arami.

**Writing – review & editing:** Nirosh Manohara.

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
