## [Decision Letter · Decision Letter 0]

2 Nov 2023

PONE-D-23-27913Coordinated human-exoskeleton locomotion emerges from regulating virtual energyPLOS ONE

Dear Dr. Arami,

Thank you for submitting your manuscript to PLOS ONE. After careful consideration, we feel that it has merit but does not fully meet PLOS ONE’s publication criteria as it currently stands. Therefore, we invite you to submit a revised version of the manuscript that addresses the points raised during the review process.

We look forward to receiving your revised manuscript.

Kind regards,

Imre Cikajlo, Ph.D.

Academic Editor

PLOS ONE

Journal Requirements:

This work is supported by NSERC Discovery, John R. Evans Leaders Fund (JELF)-Canada

Foundation for Innovation, Ontario Research Fund (ORF), and the New Frontiers in Research

Fund under Grant NFRFE-2018-01698.

- AA received NSERC Discovery under Grants RGPIN-2018-04850

-AA received New Frontiers in Research Fund - Exploration under Grants NFRFE2018-01698 and

NFRFE2022-620 

- AA received John R. Evans Leaders Fund Canadian Foundation for Innovation

- AA received Ontario Research Fund (ORF)

- HD received scholarship NSERC CGS-M

Reviewers' comments:

Reviewer's Responses to Questions

**Comments to the Author**

1. Is the manuscript technically sound, and do the data support the conclusions?

Reviewer #1: Partly

Reviewer #3: Yes

2. Has the statistical analysis been performed appropriately and rigorously? 

Reviewer #1: Yes

Reviewer #3: N/A

3. Have the authors made all data underlying the findings in their manuscript fully available?

Reviewer #1: Yes

Reviewer #3: Yes

4. Is the manuscript presented in an intelligible fashion and written in standard English?

Reviewer #1: No

Reviewer #3: Yes

5. Review Comments to the Author

Reviewer #1: This paper presents further work on the VER method. The VER allows control design in the phase space, where the control input is guided towards the limit cycle. This introduces several advantages, like time independence, coordinated motion, phase synchronization, etc.

Overall, I think the control method is very interesting and should be explored further. I also looked at the papers [38] and [39]. I believe the main flaw of the current work is the missing comparison to a “classical” controller with recorded trajectories. The controller is shown to improve the user effort compared to the passive movement and a lot of advantages are mentioned. But this does not really show that such “coordinated” controller is better than a standard, recorded trajectory one. Why is there no such comparison?

Or, alternatively, a better explanation on how one can measure the coordination between the exoskeleton and the user would help strengthen the claims.

Other major comments:

1 – Introduction: “...still far from optimal due to the unresolved human-exoskeleton coordination problem at the controller level.” -> I am not sure how the authors reached this conclusion, since there are a lot of works that currently reach very high “optimal” levels of assistance (metabolic cost reductions) using very straightforward control designs. Although, to my knowledge, these achievements were not yet generalized to different tasks. Or is this statement intended towards the current rehabilitation exoskeletons? In that case, the statement should be written differently.

2 – Introduction: “It has been shown that passive trajectory tracking controllers worsen human-exoskeleton coordination.” – there should be a citation here. And I think this is an important statement, since this paper tries to improve on this coordination problem.

3 – Figure 2 – a-d are identical from [39] and should be noted as such in the figure caption.

4 – Section 2.4 “However, to maximize the similarity of the resultant gait to natural walking, the knee and hip limit cycles are designed similar to natural walking; see Fig.2 a-d.” I understood only after looking at the [39] that the desired trajectories/limit cycles are modified to be stable. Is there any indication how this modification affects the gait?

5 - How is the “Passive” mode realized. Is the exoskeleton in a zero torque mode or is it completely turned off? Please add a sentence to clarify this. This is important, assuming that the controller is completely turned off, even having friction compensation, would undoubtedly reduce the muscle activations in the active mode.

6 – In the experimental results, the protocol is explained again. I think this is not needed, since the experimental protocol is written in section 3.1

7 – Section 4.2, Kinematics, second paragraph. Only the mass of the exoskeleton is considered to affect the gait. How about kinematic restriction imposed by the exoskeleton, like the simplified passive ankle, and DoF knee and hip?

8 – The idea about the central generator is interesting. But I would argue that more literature review in this area would be required to support this statement.

9 – Section 5, second paragraph. What is meant with the absence of conflicts?

10 – Section 5, last paragraph. “To sum up,...” sentence is strangely written, please consider revising it.

Minor comments:

- Section 2.2, line 8, there is a typo “...is almost zero, hence”

Reviewer #3: The authors should add the main conclusions of the study in their Abstract.

I recommend adding Figure S11 (Supporting material) to the content of the main text of the paper, as it illustrates the exoskeleton used, helping the understanding of Plos One readers.

Figure captions are too long. I suggest keeping only a brief presentation of the content, leaving more detailed descriptions for the main text of the paper.

6. PLOS authors have the option to publish the peer review history of their article (what does this mean?). If published, this will include your full peer review and any attached files.

Reviewer #1: No

Reviewer #2: **Yes: **Dr. Vineet Vashista, Associate Professor, IIT Gandhinagar

Reviewer #3: No

---

## [Author Response · Author response to Decision Letter 0]

20 Jan 2024

Please the response to the reviewers letter attached.

---

## [Decision Letter · Decision Letter 1]

1 Mar 2024

PONE-D-23-27913R1Coordinated human-exoskeleton locomotion emerges from regulating virtual energyPLOS ONE

Dear Dr. Arami,

Thank you for submitting your manuscript to PLOS ONE. After careful consideration, we feel that it has merit but does not fully meet PLOS ONE’s publication criteria as it currently stands. Therefore, we invite you to submit a revised version of the manuscript that addresses the points raised during the review process.

We look forward to receiving your revised manuscript.

Kind regards,

Imre Cikajlo, Ph.D.

Academic Editor

PLOS ONE

Journal Requirements:

Additional Editor Comments:

Please carefully examine the revision of the Reviewer 2 and provide point-to-point answers to resolve the remaining issues. Take in consideration also the typos addressed by the Reviewer 1.

Reviewers' comments:

Reviewer's Responses to Questions

**Comments to the Author**

1. If the authors have adequately addressed your comments raised in a previous round of review and you feel that this manuscript is now acceptable for publication, you may indicate that here to bypass the “Comments to the Author” section, enter your conflict of interest statement in the “Confidential to Editor” section, and submit your "Accept" recommendation.

Reviewer #1: All comments have been addressed

Reviewer #2: (No Response)

Reviewer #3: All comments have been addressed

2. Is the manuscript technically sound, and do the data support the conclusions?

Reviewer #1: Yes

Reviewer #2: Yes

Reviewer #3: Yes

3. Has the statistical analysis been performed appropriately and rigorously? 

Reviewer #1: Yes

Reviewer #2: Yes

Reviewer #3: Yes

4. Have the authors made all data underlying the findings in their manuscript fully available?

Reviewer #1: Yes

Reviewer #2: Yes

Reviewer #3: Yes

5. Is the manuscript presented in an intelligible fashion and written in standard English?

Reviewer #1: No

Reviewer #2: Yes

Reviewer #3: Yes

6. Review Comments to the Author

Reviewer #1: All my major comments were addressed.

I noticed that there are many typos in the paper. I would suggest the authors read the paper again and correct them.

Here are some that I found:

- “by” and “has” in line “Given gait variability, which can be intensified by an impairment, intention decoding based on gait kinematics has not been successful so far [33-35] despite the recent achievements in real-time gait phase estimation [36-39].”

- “Straighten” in line “Accordingly, we removed the flexion behavior of the knee joint during the stance phase; hence, the controller encourages the participants to straighten their leg during the stance.”

- “Experiment” in “The testing sessions were different for each experiment.”

- “were” in “Nine participants (7 male and 2 female, age: 23.9 ± 3.2 years, body mass: 72.3 ± 7.0 kg, height: 177 ± 6.3 cm; mean±standard deviation) participated in the Experiment 1.”

Reviewer #2: The paper discusses an interesting case of using VER. The authors have earlier worked on the topic and have extended it here with human experimentation. The manuscript is written well and present a thought out study. However, there are a few points that need more work.

- The authors provide a reference to their earlier works in the introduction to establish the contribution of the current paper. It appears that the concept has been developed earlier and a feasibility one human study has been published. Thus, the main contribution of the current work is implementation of the same over multiple subjects (n=9). This makes the current contribution of the paper weaker as only 9 subjects were tested, and no disabled walking studies were conducted. Also, it is not clear how the experiments 1 and 2 design strengthen the point that such approach can be useful with disabled walking.

- The second major comment is that the discussion section is very weak. Considering that the introduction section introduces the readers to the various strategies being tested in the community working on human in the loop, adaptive controllers, etc. the discussion section does not establish the proposed uniqueness and advantages as claimed for this work. This needs to be addressed.

- A claim "that passive trajectory tracking controllers worsen human-exoskeleton coordination" is made using reference [15] - this is an old reference. Are there newer references supporting this claim?

- Section 2.4 puts "Lack of actuation at the ankle joint in our exoskeleton results in walking pattern slightly different than natural walking. ... natural walking, the knee and hip limit cycles are designed similar to natural walking ..." This needs to be discussed further as it is not clear how this impacts the performance and how this will be taken care of disabled walking.

Reviewer #3: The authors responded to all comments satisfactorily. The work presents investigations relevant to the literature in the area of human gait analysis.

7. PLOS authors have the option to publish the peer review history of their article (what does this mean?). If published, this will include your full peer review and any attached files.

Reviewer #1: No

Reviewer #2: No

Reviewer #3: No

---

## [Author Response · Author response to Decision Letter 1]

26 Mar 2024

(Please see the uploaded response letter)

The authors would like to thank the Associate Editor and Reviewers for their valuable comments on our manuscript, Submission ID PONE-D-23-27913R1, entitled “Coordinated human-exoskeleton locomotion emerges from regulating virtual energy”. In the following, you may find our responses to the questions and comments of the reviewers. We addressed all the reviewers’ comments one by one in this response letter and revised the manuscript accordingly. Changes are highlighted in the attached copy of the revised manuscript at the end of this document to facilitate finding the relevant modifications to each comment. The highlighted captions also show the revised captions and/or figures.

Best Regards,

Rezvan Nasiri, Hannah Dinovitzer, Nirosh Manohara, and Arash Arami

University of Waterloo

Journal Requirements

Response: The following citations were added due to the reviewers’ request.

• Chen, B., Ma, H., Qin, L. Y., Gao, F., Chan, K. M., Law, S. W., ... & Liao, W. H. (2016). Recent developments and challenges of lower extremity exoskeletons. Journal of Orthopaedic Translation, 5, 26-37.

• Lee, H., Ferguson, P. W., & Rosen, J. (2020). Lower limb exoskeleton systems—overview. Wearable Robotics, 207-229.

Additional Editor Comments

Please carefully examine the revision of the Reviewer 2 and provide point-to-point answers to resolve the remaining issues. Take in consideration also the typos addressed by the Reviewer 1.

Response: We addressed all the reviewers’ points carefully, and revised the paper accordingly.

Reviewer 1

1. I noticed that there are many typos in the paper. I would suggest the authors read the paper again and correct them. Here are some that I found:

“by” and “has” in line “Given gait variability, which can be intensified by an impairment, intention decoding based on gait kinematics has not been successful so far [33-35] despite the recent achievements in real-time gait phase estimation [36-39].”

“Straighten” in line “Accordingly, we removed the flexion behavior of the knee joint during the stance phase; hence, the controller encourages the participants to straighten their leg during the stance.”

“Experiment” in “The testing sessions were different for each experiment.”

“were” in “Nine participants (7 male and 2 female, age: 23.9 ± 3.2 years, body mass: 72.3 ± 7.0 kg, height: 177 ± 6.3 cm; mean±standard deviation) participated in the Experiment 1.”

Response: We like to thank the reviewer for pointing out those issues; we now corrected all the typos. In addition, we proofread the paper carefully to rectify any English issues.

Reviewer 2

The paper discusses an interesting case of using VER. The authors have earlier worked on the topic and have extended it here with human experimentation. The manuscript is written well and present a thought out study. However, there are a few points that need more work.

2. The authors provide a reference to their earlier works in the introduction to establish the contribution of the current paper. It appears that the concept has been developed earlier and a feasibility one human study has been published. Thus, the main contribution of the current work is implementation of the same over multiple subjects (n=9). This makes the current contribution of the paper weaker as only 9 subjects were tested, and no disabled walking studies were conducted. 

Response: Thanks for this comment. Please note that, our previous work included a theoretical and simulation work (referring to RA-L paper) and a conference paper with only one participant with a different focus which was the ability of VER in estimating gait phase (a relatively short conference paper). That one participant experiment was only a proof of concept and did neither follow similar experimental protocol or data collection and analysis, for instance it did not include any muscle activity (EMG) data. 

The main contributions of the presented paper are: (1) experimenting the controller for the first time on a group of 9 able-bodied individuals, (2) vast experimental data analysis in terms of kinematics, controller command, questionnaire, and muscle activity of the users and comparing the VER resultant gait with natural walking, (3) comparing the controller with an effective controller presented in the filed in terms of kinematic analysis and muscle EMG reduction, (4) presenting a new metric to measure the individuals contribution to the gait as well as presenting the analytical convergence proof by considering the user active contribution to the gait, and hypnotizing that human lower limb neuromuscular system works similar to the VER controller. We believe that the provided points make the contributions of this paper solid and distinct. Moreover, we clearly mentioned in the discussion (now changed its name to Conclusion as the discussions are merged to the Results and Discussion section) that the presented controller is not applied on motor-impaired individuals, and it is considered as our future work. According to reviewer’s comment, we revised the Abstract to clarify the contributions of this work; please check the Abstract of the revised Conclusion.

3. Also, it is not clear how the experiments 1 and 2 design strengthen the point that such approach can be useful with disabled walking.

Response: Thanks for this comment. Our controller is not aiming to help individuals who cannot contribute to the motion at all, but those with residual ability to move (for instance, individuals with incomplete spinal cord injury or stroke). The main challenge for those individuals with various motor impairments is active contribution to motion to regain their ability to walk again and the main drawback of the existing controllers is the point that the lower limb exoskeleton controllers mainly cannot coordinate with human gait variability. Accordingly, this paper focused on analyzing the gait naturalness and coordination performance of the resultant gait by VER controller. Experiment 1 is designed to study the VER performance in a group of 9 healthy individuals. In this experiment, the treadmill speed is selected similar to the walking speed suggested for individuals with motor impairments [22-30] to test it closer to the needs of those individuals. Having said that we acknowledge that this work is a step needed to build the foundation for next studies on people with those motor disabilities. This is now further clarified in the revised Conclusion section.

Experiment 2 is also designed to compare our controller with the path controller [22-24], which is a well-known and effective controller in the rehabilitation field and was experimented on a wide range of individuals with motor impairment [22,23], in terms of joint kinematics and muscle EMG reduction. Also note that this experiment was added to this work as suggested by another reviewer in the previous round. In the revised version of the paper, we added a new paragraph at the beginning of the Experiment Design section to explain our perspectives in design of experiment 1 and experiment 2.

4. The second major comment is that the discussion section is very weak. Considering that the introduction section introduces the readers to the various strategies being tested in the community working on human in the loop, adaptive controllers, etc. the discussion section does not establish the proposed uniqueness and advantages as claimed for this work. This needs to be addressed.

Response: Thanks for this comment. We noticed that some parts of our discussion were already within the Results section. As removing those parts and adding them to the separate Discussion section could undermine the readability of this paper, we decided to join the Discussion and Results sections as Results and Discussion section. 

The revised Results and Discussion section includes the analysis and discussion of human-robot interaction with a focus on muscle activity analysis, and the analysis of kinematics, gait spatiotemporal parameters, perceived quality of interaction through questionnaire, and the analysis of naturalness of assisted gait. We also discussed the control performance of VER, and compared it to path controller equipped with a robust gait phase estimator. 

While our Introduction mentioned other controllers in the field to set the stage for the VER in this domain, the comparison of VER with various control strategies is beyond the scope of this work. This is now clarified in the Conclusion section. However, as also asked by another reviewer in the previous round we have already ran new experiments and added the results and comparison with path controller tested on three individuals, as one of the successful controllers in the gait rehabilitation field (as a reminder Experiment 2 was added later to the paper to enrich the results and conclusion). 

The Conclusion section, while mostly focused on summarizing the main experimental results and achievements of the paper, is now revised according to the reviewer’s comment to clarify the limitations and the future work. 

5. A claim "that passive trajectory tracking controllers worsen human-exoskeleton coordination" is made using reference [15] - this is an old reference. Are there newer references supporting this claim?

Response: The following paper is cited as [16] to address the reviewer’s concern.

• Chen, B., Ma, H., Qin, L. Y., Gao, F., Chan, K. M., Law, S. W., ... & Liao, W. H. (2016). Recent developments and challenges of lower extremity exoskeletons. Journal of Orthopaedic Translation, 5, 26-37. 

6. Section 2.4 puts "Lack of actuation at the ankle joint in our exoskeleton results in walking pattern slightly different than natural walking. ... natural walking, the knee and hip limit cycles are designed similar to natural walking ..." This needs to be discussed further as it is not clear how this impacts the performance and how this will be taken care of disabled walking. 

Response: Most of the existing lower limb assistive exoskeletons are using the passive ankle joints, since having an active ankle joint makes the exoskeleton bulky (impractical for those users with residual ability to move) and expensive [22], please also see the review paper below, which is now cited as [44]. In addition, in controller failure cases, active ankles may lead to a fall or major injury for the individuals with motor impairment. Accordingly, considering the cost efficiency, applicability, and safety limitations, our lower limb assistive exoskeleton also utilizes the passive ankle joints. Hence, only hip and knee joints are left to be controlled. 

In most of the existing controllers for lower limb assistive exoskeletons, the knee and hip reference motions are designed similar to human normal walking and may be slightly modified to address the exoskeleton and controller limitations. Accordingly, the trajectories are similar but not identical to able-bodied individual’s motion profiles; please check the following review paper. 

The same scenario was also considered in the design of the reference trajectory leading to the limit cycles for hip and knee in our controller. We clarified these points in the revised version of Mathematics section; please check section 2.4 in the revised paper. The mentioned review paper was also added to the paper. 

• Lee, H., Ferguson, P. W., & Rosen, J. (2020). Lower limb exoskeleton systems—overview. Wearable Robotics, 207-229.

---

## [Editor Report · Decision Letter 2]

28 May 2024

Coordinated human-exoskeleton locomotion emerges from regulating virtual energy

PONE-D-23-27913R2

Dear Dr. Arami,

We’re pleased to inform you that your manuscript has been judged scientifically suitable for publication and will be formally accepted for publication once it meets all outstanding technical requirements.

Kind regards,

Imre Cikajlo, Ph.D.

Academic Editor

PLOS ONE
---

## [Editor Report · Acceptance letter]

20 Sep 2024

PONE-D-23-27913R2 

PLOS ONE

Dear Dr. Arami, 

I'm pleased to inform you that your manuscript has been deemed suitable for publication in PLOS ONE. Congratulations! Your manuscript is now being handed over to our production team.

Kind regards, 

on behalf of

Professor Imre Cikajlo 

Academic Editor

PLOS ONE